# Lysine: Sources, Metabolism, Physiological Importance, and Use as a Supplement

**DOI:** 10.3390/ijms26188791

**Published:** 2025-09-09

**Authors:** Milan Holeček

**Affiliations:** Department of Physiology, Faculty of Medicine, Charles University, 500 03 Hradec Králové, Czech Republic; holecek@lfhk.cuni.cz

**Keywords:** carnitine, desmosine, homoarginine, homocitrulline, lysine–arginine antagonism, cadaverine, saccharopine

## Abstract

This article provides a comprehensive review and explores the gaps in current knowledge of lysine metabolism in humans and its potential nutritional and therapeutic indications. The first part of this study examines lysine sources, requirements, transport through the plasma membrane, lysine catabolism, and its disorders. The central part is focused on post-translational modifications of lysine in proteins, primarily desmosine formation in elastin, hydroxylation in collagen, covalent bonds with glutamine, methylation, ubiquitination, sumoylation, neddylation, acylation, lactylation, carbamylation, and glycation. Special sections are devoted to using lysine as a substrate for homoarginine and carnitine synthesis and in nutrition and medicine. It is concluded that the identification and detailed knowledge of writers, readers, and erasers of specific post-translational modifications of lysine residues in proteins is needed for a better understanding of the role of lysine in epigenetic regulation. Further research is required to explore the influence of lysine availability on homoarginine formation and how the phenomenon of lysine–arginine antagonism can be used to influence immune and cardiovascular functions and cancer development. Of unique importance is the investigation of the use of lysine in osteoporosis therapy and in reducing the resorption of harmful substances in the kidneys, as well as the therapeutic potential of polylysine and lysine analogs.

## 1. Introduction

Lysine (2,6-diaminohexanoic acid, symbols Lys or K) is a basic, proteinogenic, ketogenic, and nutritionally essential amino acid.



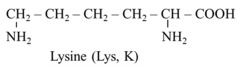



More than 60 years ago, it was proven that lysine is the first limiting amino acid in cereals, and using the nitrogen balance technique, it was shown in children that the addition of lysine to a basal wheat diet markedly increases nitrogen retention [1]. Therefore, lysine has been extensively used to fortify foods and as a dietary supplement, particularly in poor countries where cereal-based diets are the main source of nutrients [2]. Recently, the role of post-translational modification (PTM) of lysine residues in proteins has become a focus of interest. For instance, hydroxylysine is an essential component of collagen, and desmosine, formed from allysine, connects tropoelastin molecules and ensures elastin’s flexible and stable structure. Many publications deal with carnitine, which is formed from trimethyllysine released during protein breakdown. The so-called “readers” of PTMs play a crucial role in the regulation of metabolism and epigenetics, i.e., the control of gene expression. Exciting is the phenomenon of lysine–arginine antagonism and its potential application in influencing NO production, tissue perfusion, immunity, and carcinogenesis [3,4]. Lysine analogues are used in the prevention and therapy of bleeding and investigated for the therapy and prevention of viral diseases [5,6,7].

This article aims to provide an overview of lysine’s physiological importance and its possible use as a dietary supplement and in therapy. We hope that a combination of scientific, clinical, and nutritional information will be of interest to clinicians and researchers in the fields of nutrition and amino acid and protein metabolism.

## 2. Sources and Requirements

Lysine is abundantly present in foods of animal origin. Plant sources with a high lysine content include legumes, nuts, and dry fruit. The small lysine content limits the nutritional value of cereals, rice, and corn [8,9]. The availability of lysine in a diet can decrease food processing. At high temperatures, lysine undergoes a Maillard reaction in the presence of carbonyl compounds, resulting in the formation of glycated lysine, whose metabolic use is questionable [10]. Isotope tracer studies performed in humans have shown that the gut microbiota is also a source of lysine for the host. However, the quantitative contribution of microbial lysine cannot be reliably estimated and could be less than the contribution of other amino acids [11,12,13].

The required daily supply of lysine for adults is ~30 mg/kg, 40 mg/kg for children, and up to 130 mg/kg for infants [14,15,16]. The usual daily lysine intake in the Western diet is 3–7 g, 40–180 mg/kg [17]. Like threonine, lysine does not undergo transamination and cannot be synthesized from the corresponding ketoacid.

## 3. Lysine Transporters

Cationic amino acid transporters, referred to as y^+^, CAT1-4, and SLC7A1-4, enable the transport of lysine, arginine, and ornithine through the plasma membrane by facilitated diffusion, and are considered the major pathway of entry and efflux of lysine and arginine in non-epithelial cells. The driving force for bidirectional transport is believed to be the electrochemical gradient of the transported compounds [18]. Cation uptake is energetically preferred in cells with a normal inside-negative membrane potential, resulting in up to five times higher concentrations of lysine in tissues when compared with its plasma concentration, ~190 μmol/L [19,20]. CAT1 is ubiquitous except for the liver; CAT2 is found in the liver, skeletal muscle, and pancreas; CAT3 is in the thymus, ovary, testis, and neurons; and CAT4 is in the brain, testes, and placenta [21].

Na^+^-independent exchangers formed by light (SLC7) and heavy (rBAT or 4F2hc) subunits are crucial for intestinal and renal absorption of cationic amino acids, including lysine, arginine, and ornithine, in exchange for neutral amino acids. Hence, unlike transport of most other amino acids, where Na^+^-dependent transporters play the dominant role in the apical side and facilitated diffusion occurs through the basolateral side, cationic amino acids use antiporters in both sides of enterocytes and cells of proximal tubules in the renal cortex [22,23]. SLC7A9 formed by SLC7 and rBAT, ensures the influx of cationic amino acids and cystine through the apical membrane in exchange for neutral amino acids. A tandem with Na^+^-dependent transporter B^0,+^AT1 (SLC6A14) enables this form of transport of cationic amino acids into the cell. SLC7A7, formed by SLC7 and 4F2hc, ensures the efflux of cationic amino acids through the basolateral membrane. Certain amounts of lysine are resorbed as part of peptides by the peptide transporter (PEPT), which is coupled with Na^+^/H^+^ exchanger (Figure 1).

### 3.1. Lysinuric Protein Intolerance

Lysinuric protein intolerance is the main hereditary disorder of lysine transport through the plasma membrane, characterized by impaired absorption of arginine, lysine, and ornithine (“ALO group”) in the small intestine and proximal tubules in the kidneys due to SLC7A7 (y^+^LAT1) mutation. The symptoms include decreased arginine, lysine, and ornithine concentrations in plasma but their elevated concentrations in urine, protein intolerance, growth retardation, hyperammonemia, hepatosplenomegaly, osteoporosis, and brain dysfunction [24]. The treatment consists of a protein-restricted diet, removing excess ammonia (e.g., by sodium phenylbutyrate or sodium benzoate), and supplementation with citrulline, which can be converted to arginine and ornithine in the liver. The effect of oral lysine administration is not clear due to intestinal malabsorption.

### 3.2. Lysine–Arginine Antagonism

Because lysine transporters through the plasma membrane also have an affinity for arginine (most ornithine in the body is synthesized from arginine and glutamate by arginase and ornithine aminotransferase, respectively), these amino acids compete with each other to bind to the carrier. Therefore, an excessive supply of one amino acid can lead to a deterioration in the transport and metabolism of both and affect the production of various compounds such as NO, ornithine, creatine, and polyamines. For instance, an arginine-enriched diet increased postprandial concentrations of arginine and ornithine (due to its formation from arginine by arginase) and decreased lysine in plasma in rats [20]. Conversely, an excess of lysine reduced the entry of arginine into cells and NO production [25]. Potential medical applications of the phenomenon of lysine–arginine antagonism are listed in Section 7.

## 4. Lysine Catabolism

The major site of lysine catabolism in humans is the mitochondria in the liver; the muscle, kidneys, heart, brain, and intestinal epithelial cells have less significant capacity [26]. The main degradation route is via saccharopine; the alternative route is the pipecolic acid pathway (Figure 2)**.**

### 4.1. The Saccharopine Pathway

The saccharopine pathway has been described in detail in several articles [13,27]. Briefly, after entering the mitochondria via a carrier for basic amino acids (SLC25A29), 2-aminoadipic acid semialdehyde (AASA) synthase, a bifunctional enzyme with domains for lysine–ketoglutarate reductase and saccharopine dehydrogenase, catalyzes the gradual conversion of lysine to saccharopine and AASA (allysine). Allysine is oxidized using AASA dehydrogenase (antiquitin) to form 2-aminoadipate, which is, through a sequence of reactions shared with the tryptophan degradation pathway, converted into two molecules of acetyl-CoA. Hence, lysine is a purely ketogenic amino acid.

Blemmings et al. [28] demonstrated in rat liver homogenates that increasing protein consumption increases the activity of lysine–ketoglutarate reductase more than lysine oxidation and lysine transport into mitochondria. Therefore, the authors concluded that lysine uptake by mitochondria was the rate-limiting step of lysine oxidation. The main role in lysine uptake by mitochondria is played by AA^+^ carrier (SLC25A29), which transports primarily lysine and arginine, less ornithine and histidine. Affinity for lysine and arginine also has ornithine carriers SLC25A15 (ornithine carrier 1) and SLC25A2 (ornithine carrier 2). SLC25A15 catalyzes the citrulline/ornithine exchange across the mitochondrial inner membrane required for the urea cycle. The physiological role of SLC25A2, which also has an affinity for homoarginine, is unclear [29].

### 4.2. The Pipecolate Pathway

Lysine catabolism via the pipecolic acid pathway, which converges with the saccharopine pathway at the point of AASA, is active in the mammalian brain [30]. However, it is not fully characterized at the molecular level, and its contribution to lysine metabolism is unclear [27].

The first step, the formation of 2-oxo-6-aminocaproate, which is spontaneously converted to piperideine-2-carboxylate (P2C), results from lysine transamination or deamination. The enzyme ketimine reductase, called µ-crystallin, encoded by the CRYM gene in humans, converts P2C to pipecolic acid. Pipecolate is further converted by pipecolic acid oxidase into piperideine-6-carboxylate (P6C), which can be spontaneously converted to AASA and enter the saccharopine pathway. Because P6C is in equilibrium with AASA and pyrroline-5-carboxylate reductase can convert P6C back to pipecolic acid, there is a possibility of reversible conversion of AASA and pipecolate [27,31].

In the spotlight of neurochemists is ketimine reductase (µ-crystallin), which acts as a thyroid hormone-binding protein. The binding of T3 strongly inhibits the enzyme and the flux of lysine through the pipecolic acid pathway. The regulation of µ-crystallin by thyroid hormones probably has an important role in neuronal development and function [32]. Altered expression or mutations of CRYM are associated with psychiatric and neurological diseases, including Huntington’s disease, amyotrophic lateral sclerosis, and deafness [33].

### 4.3. Disorders of Lysine Catabolism

The disorders of lysine catabolism requiring early implementation of lysine-restricted diets are glutaric aciduria type 1 and pyridoxine-dependent epilepsy. Hyperlysinemia, saccharopinuria, and 2-aminoadipic and 2-oxoadipic aciduria are generally considered benign disorders [27,34]:*Glutaric aciduria type 1* is a rare autosomal recessive disease caused by glutaryl-CoA dehydrogenase deficiency. Lysine and tryptophan (which is also catabolized via glutaryl-CoA) and by-products of glutaryl-CoA metabolism, such as glutaric acid, 3-hydroxyglutaric acid, and glutarylcarnitine, accumulate in the body, primarily in the brain. Therapy aims to prevent brain injury using lysine-free, arginine-fortified amino acid supplements [35].*Pyridoxine-dependent epilepsy (antiquitin deficiency)* is caused by a mutation in AASA dehydrogenase (antiquitin). There is an accumulation of allysine, pipecolic acid, and P6C, which forms inactive complexes with pyridoxal phosphate (vitamin B_6_) acting as a cofactor of several enzymes, primarily decarboxylases and aminotransferases [36]. The typical clinical picture includes seizures unresponsive to conventional antiepileptic drugs. High doses of pyridoxine, lysine-restricted diets, and the supply of arginine, which competes with lysine for intestinal absorption, are used for therapy [37]. A new treatment option for reducing the level of neurotoxic AASA is the inhibition of AASA synthase [38].*Hyperlysinemia* is an autosomal recessive disorder characterized by increased lysine levels in the blood caused by a defect of the bifunctional AASA synthase.*Saccharopinuria* is caused by a mutation in the saccharopine dehydrogenase domain of the AASA synthase gene, characterized by high levels of saccharopine in urine. Although saccharopinuria is generally considered benign, there are reports indicating impaired function of the urea cycle, hyperammonemia, hypercitrullinemia, intellectual impairment, and neurological problems [39]. Some alterations are probably due to the inhibitory influence of lysine and saccharopine on urea cycle enzymes, argininosuccinate synthetase, argininosuccinate lyase, and arginase [40]. In *Caenorhabditis elegans*, mutations in the saccharopine dehydrogenase domain of the AASA caused greatly elevated levels of saccharopine, mitochondrial damage, and reduced worm growth [41].*2-aminoadipic and 2-oxoadipic aciduria* is due to a mutation in the E1 component of 2-oxoadipic acid dehydrogenase complex. The disorder is mostly asymptomatic [27].

## 5. Physiological Importance

Lysine is a proteinogenic amino acid that acts as a substrate for post-translational modifications and the synthesis of several physiologically important substances (Figure 3).

### 5.1. Lysine in Proteins

The content of L-lysine in most proteins is 6–8%. High amounts (even over 20%) are found along with arginine in histones, basic proteins in the cell nucleus [42]. The ε-amino group of lysine possesses lone-pair electrons and, at the physiological pH (7.4), exists in the polar, protonated form. Therefore, in proteins, lysine residues are positioned on the surface or near solvent-exposed areas, where they play numerous roles in protein functions and easily participate in diverse metabolic interactions. For instance, lysine residues are a frequent part of the catalytic domain of many enzymes and enable the connection of histones with DNA and the binding of lysine-rich regions of LDL particles to LDL receptors [43,44]. Of unique clinical importance is the role of lysine residues in fibrinolysis. Lysine binding sites of plasminogen are essential for binding to fibrin, a major component of the blood clot. Upon binding to the fibrin molecule, plasminogen is activated to plasmin, a proteolytic enzyme that dissolves fibrin clots. Hence, lysine analogs, such as tranexamic acid, are used as antifibrinolytic substances [5,6] (see Section 6.9).

### 5.2. Post-Translational Modifications

Post-translational modification (PTM) is one of the body’s major mechanisms to increase the number of protein species and the diversity of their functions. So-called writers, erasers, and readers mediate the control of PTM and its impact on metabolism. Writers ensure additions of various chemical groups to the target, e.g., methylation of the ε-amino group of lysine residue by specific methyltransferase. Some PTMs, such as glycation, occur spontaneously. Readers include specialized domains of some proteins that recognize those modifications and initiate a specific metabolic response. Erasers are enzymes that are efficient in removing these chemical tags. Dysregulation of PTM is associated with several diseases, including carcinogenesis, diabetes, and neuropsychiatric disorders [45,46,47,48].

The common modifications of lysine residues include the formation of desmosine and isodesmosine in elastin, hydroxylation in collagen, covalent bonds with glutamine side chains, ubiquitination, sumoylation, neddylation, lactylation, methylation, acylation (particularly acetylation), carbamylation, and glycation.

#### 5.2.1. Desmosine and Isodesmosine Formation in Elastin

Elastin is an extracellular matrix protein organized into elastic fibers that provide elasticity, e.g., of the skin, lungs, aorta, ligaments, and urinary bladder. The soluble precursor of elastin is tropoelastin, secreted by elastogenic cells such as fibroblasts, lung alveolar cells, smooth muscle cells, and chondrocytes, which aggregates via a process known as coacervation [49]. The oxidative deamination of lysine residues by lysyl oxidase to allysine is the first step in forming crosslinks (Figure 4). The reactive aldehyde group of allysine condensates with lysine residue to form dehydrolysinonorleucine or with another allysine residue to allysine aldol, the simplest crosslinks in elastin, which are also found in collagen (next section). Condensation of three allysine molecules with one lysine creates a cyclic structure (desmosine or isodesmosine), which connects tropoelastin molecules and creates a flexible and stable structure that enables the elasticity of tissues in which elastin is present [50].

When elastin breaks down, desmosine and isodesmosine are released into the blood and excreted in the urine. Urinary desmosine excretion is a biomarker of elastin degradation and tissue injury [51]. Recent studies have revealed that dysregulation of lysyl oxidase activity is associated with multiple pathophysiological processes, including the pathogenesis of atherosclerosis [52].

#### 5.2.2. Lysine Hydroxylation

Hydroxylysine is formed primarily from lysine present in collagen, the chief structural protein of mammalian skin and connective tissue, using lysyl hydroxylase (Figure 5):

Certain hydroxylysine residues in collagen molecules form chemical bonds with sugars, primarily galactose and glucosyl-galactose, thus contributing to collagen’s toughness and resiliency. Another function of hydroxylysine residues is participating in the crosslinking of collagen molecules via covalent bonds. The formation is catalyzed by lysyl oxidase, which deaminates the lysine or hydroxylysine residues to the reactive aldehydes allysine or hydroxyallysine, respectively. These then react without the participation of enzymes with a nearby lysine or hydroxylysine ε-amino group to form intermolecular links (previous section). This crosslinking is essential for the biomechanical properties of collagen and bone mineralization, which starts in gaps adjacent to crosslinks [53].

Hydroxylysine released during collagen degradation cannot be reused for protein synthesis, and it is catabolized via the allysine pathway or excreted in the urine. Hence, urinary excretion of hydroxylysine and its glycosides can be used along with hydroxyproline as an index of collagen degradation [54,55].

#### 5.2.3. Formation of Covalent Bonds Between Glutamine and Lysine Residues

The ε-amino group of lysine residues is a substrate for transglutaminases, which catalyze the formation of covalent bonds and crosslinks, primarily with amide groups of glutamine residues (Figure 6). The proteins formed in this way are resistant to mechanical and chemical influences. For instance, transglutaminases play a role in hemostasis, apoptosis, and tissue healing. Upregulated transglutaminases play a critical role in liver and kidney fibrosis progression via stabilizing extracellular matrix proteins and activating TGF-β [56].

#### 5.2.4. Lysine Methylation

Lysine methylation and demethylation by protein lysine methyltransferases ("writers") and demethylases ("erasers") form mono-, di-, and trimethyl lysine residues. Methylated lysine residues in histones recognized by distinct effector proteins ("readers") in a manner that depends on the neighboring amino acid sequence and methylation state play a crucial role in epigenetic regulation of gene expression and contribute to various biological processes [46,57]. Numerous studies have indicated that methylated lysine residues prevent ubiquitination and, therefore, influence proteolysis and protein turnover [58]. Dysregulation of methylation is linked to various diseases, including cancer, inflammation, and genetic disorders [45,46]. Lysine methylation, primarily in muscle proteins, is the initial step in carnitine synthesis (Section 5.4).

#### 5.2.5. Lysine Ubiquitination

Ubiquitination is the term used for the attachment of a small regulatory protein called ubiquitin to a protein molecule, primarily on the ε-amino group of lysine residues. The "writer" is a complex of three enzymes: ubiquitin-activating enzyme (E1), ubiquitin-conjugating enzyme (E2), and ubiquitin ligase (E3). A polyubiquitin chain attached to lysine residue labels the proteins intended for ATP-dependent proteolysis in proteasomes and stimulates various downstream signals involved, such as apoptosis, cell cycle control, NF-κB activation, and DNA repair [58]. A crucial role in the identification of targeted proteins and proteasome activation is played by ubiquitin ligases (E3). These include the genes designated MuRF1 (Muscle Ring Finger 1) and atrogin-1, also known as Muscle Atrophy F-box (MAFbx), which are involved in the pathogenesis of cachexia in various muscle-wasting disorders [59].

#### 5.2.6. Lysine Sumoylation

Sumoylation is characterized by a bond of a small ubiquitin-like modifier (SUMO) protein to lysine residues within specific motifs of many substrates. Four SUMO isoforms have been identified. Like ubiquitination, sumoylation is a reversible process that involves three classes of enzymes and can be reversed by specific proteases. Sumoylation is involved in various cellular processes, including DNA damage repair, cell cycle progression, apoptosis, immune response, and epigenetic induction or repression of certain genes. Unlike ubiquitination, sumoylation is not used to tag proteins for degradation. The dysregulation of the SUMO system is associated with many diseases, particularly cancer and neurodegeneration [60,61].

#### 5.2.7. Lysine Neddylation

Neddylation is a reversible binding of a ubiquitin-like protein termed neural precursor cell-expressed developmentally downregulated 8 (NEDD8) to the lysine residue of the substrate protein. Like ubiquitination and sumoylation, neddylation is catalyzed by the enzyme cascade, namely, NEDD8 activating enzyme (E1), NEDD8 conjugating enzyme (E2), and NEDD8 ligase (E3). Deneddylation by specific deneddylases releases free NEDD8 for another cycle of neddylation. Neddylation substrates are classified into cullins and non-cullin proteins. Cullins support ubiquitin ligases (E3) and play a role in protein degradation. Non-cullin substrates include, for example, oncogenes, ribosomal proteins, histones, and transcription factors. The neddylation pathway is abnormally activated in various diseases, such as cancer, neurodegenerative disorders, cardiomyopathies, atherosclerosis, and chronic liver diseases [62,63].

#### 5.2.8. Lysine Acylation

According to the differences in hydrocarbon chain length, lysine acylation includes acetylation, propionylation, butyrylation, succinylation, malonylation, glutarylation, crotonylation, etc. Acylation is sensitive to the metabolic state of the cell, primarily the level of a particular acyl-CoA. Lysine acetylation has been widely studied.

Reversible acetylation of the epsilon amino group of the side chain lysine residues is controlled by lysine acetyltransferases (KATs), the "writers", which use acetyl-CoA as a substrate. In the mitochondria, lysine acetylation can also occur nonenzymatically. Deacetylation is achieved by deacetylases (KDACs), the "erasers", which include Zn^2+^-dependent lysine deacetylases and sirtuins, a group of NAD^+^-dependent deacetylases (Figure 7).

Acetylation neutralizes the positively charged lysine residues and blocks other lysine modifications, such as ubiquitination. Like methylation, lysine acetylation/deacetylation has been extensively studied in histones. Bromodomain-containing proteins, "readers" of lysine acetylation, can recognize acetylated lysine residues and influence gene transcription and transduce the signal carried by acetylated lysine residues into various normal or abnormal phenotypes [47]. Dysregulation of KATs, KDACs, and bromodomain-containing proteins plays a role in aging and pathogenesis of cancer and neurodegenerative diseases, such as Parkinson’s disease [47,48]. Recently, lysine acetylation of actin was revealed to regulate cytoplasmic actin polymerization and intracellular calcium transfer [64].

#### 5.2.9. Lysine Lactylation

Lysine lactylation involves the covalent linkage of lactate to lysine residues in proteins, primarily histones, induced by lactate accumulation. Histone lysine lactylation alters chromatin spatial configuration and the expression of associated genes. Lactylation plays a significant role in immune regulation, cell cycle, vascular function, and tumor development [65].

#### 5.2.10. Lysine Carbamylation

Carbamylation or carbamoylation refers to a reaction of cyanates (NCO^−^), originating from the breakdown of urea or due to inflammation from environmentally derived thiocyanate, with amino nitrogen to form a carbamoyl group (-CONH_2_). Carbamylation of ε-amino nitrogen of lysine residues produces homocitrulline, an amino acid that is one methylene group longer than citrulline. The reaction occurs both with free lysine and lysine residues in proteins (Figure 8).

Increased carbamylation has been linked to complications of chronic renal diseases, including renal fibrogenesis, cataracts, and rheumatoid arthritis [66,67]. The carbamylation of LDL-type lipoprotein particles leads to their impaired uptake by LDL receptors from the blood and atherosclerosis [44]. An indicator of the rate of carbamylation is the level of carbamylated albumin and hemoglobin, as well as protein-bound homocitrulline [67]. Up to now, no studies have evaluated the effect of removing carbamylated substances, for example, as part of dialysis, on the progression of renal disease.

#### 5.2.11. Glycation

In glycation, the aldehyde group of a reducing sugar binds nonenzymatically to a free amino group of a protein, primarily lysine and arginine residues, initiating a series of changes known as Maillard reactions. The consequence is the crosslinking of protein segments, resulting in altered conformation and protein function. The interaction of glycation and the influence of reactive oxygen species leads to the formation of substances referred to as advanced glycation end-products (AGEs), such as pentosidine and glyoxal lysine, crosslinking of protein segments, altered function of the protein, and possible tissue damage. For instance, glycation of the collagen of the basal membranes of glomeruli leads to the development of glomerulosclerosis in patients with diabetes [68].

### 5.3. Lysine as a Substrate for Homoarginine Synthesis

Homoarginine is a non-proteinogenic amino acid that differs from arginine by an additional methylene (-CH_2_-) group. It is synthesized in several tissues, including the liver, kidneys, muscle, adipose tissue, and small intestine [69]. There are two routes of homoarginine synthesis from lysine (Figure 9):The main pathway is via arginine:glycine amidinotransferase. When lysine is the substrate instead of glycine, the enzyme does not catalyze the synthesis of guanidinoacetate (Arg + Gly → Orn + guanidinoacetate), the precursor of creatine, but the synthesis of homoarginine (Arg + Lys → Orn + homoarginine).A less significant pathway is via argininosuccinate synthetase and argininosuccinate lyase (enzymes of the urea cycle) from homocitrulline formed from lysine by ornithine carbamoyltransferase when lysine is used as a substrate instead of ornithine or by the reaction of lysine with cyanates (carbamylation), especially in uremia [70,71].

**Figure 9 ijms-26-08791-f009:**
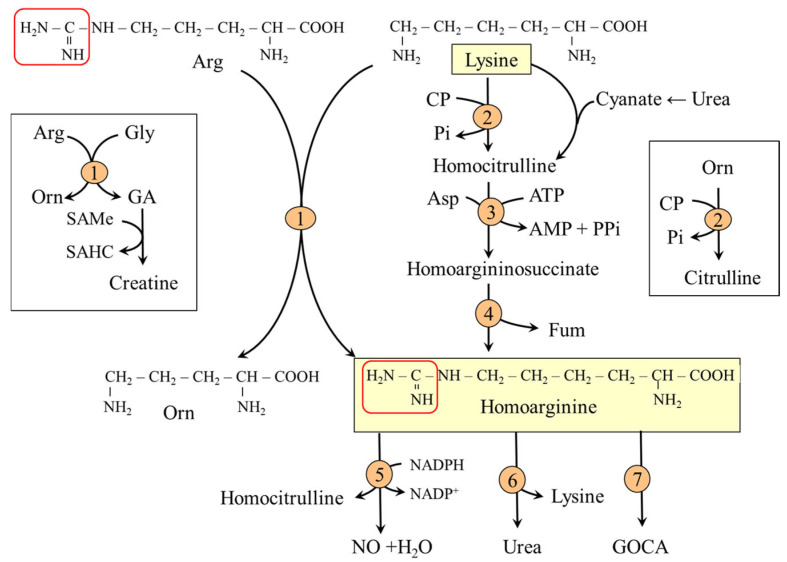
Homoarginine synthesis from lysine and potential pathways of its metabolism. 1, arginine:glycine amidinotransferase (AGAT); 2, ornithine carbamoyltransferase (OCT); 3, argininosuccinate synthetase; 4, argininosuccinate lyase; 5, NO synthase; 6, arginase; 7, alanine–glyoxylate aminotransferase. The box on the left shows a diagram of the role of AGAT in the synthesis of GA; the box on the right shows the role of OCT in citrulline synthesis. Abbreviations: CP, carbamoyl phosphate; GA, guanidinoacetate; GOCA, 6-guanidino-2-oxocaproic acid.

Homoarginine serves as an alternative substrate of arginine for all three isoforms of NO synthase and, via N^ω^-hydroxy-L-homoarginine, an intermediate formed during NO synthase reaction, as a competitive inhibitor of arginase. Unclear is the possible role of homoarginine as an alternative substrate for arginase to form urea and lysine and its degradation to 6-guanidino-2-oxocaproic acid (GOCA) by alanine–glyoxylate aminotransferase [72,73,74].

Homoarginine concentration in human plasma ranges between 1 and 2 µmol/L. Several studies have demonstrated that a decrease in homoarginine concentration in diseases of the kidneys and cardiovascular system is associated with myocardial dysfunction and an increased risk of cardiovascular events, such as stroke [68,70]. Although the pathophysiological mechanism of the adverse effects of homoarginine deficit has not been elucidated, homoarginine supplementation has been investigated as a strategy to influence the development of cardiovascular diseases [72,73,75,76].

### 5.4. Lysine and Carnitine Synthesis

Carnitine (β-hydroxy-trimethylammonium butyrate) is essential for the transport of higher fatty acids into the mitochondria to be oxidized (Figure 10). Carnitine synthesis is initiated by post-translational methylation of lysine residues, primarily in muscle proteins, using protein lysine methyltransferase (‘writer’) and SAMe to form N-trimethyllysine (TML), which is released during protein degradation. Enzymes for converting TML to butyrobetaine are expressed in most tissues, including the liver, kidneys, muscle, heart, and brain. The last enzyme in carnitine synthesis (butyrobetaine dioxygenase) is expressed in the liver, kidneys, and brain [77]. Carnitine released into the circulation is transported to tissues by OCTN2 (organic cation/carnitine transporter novel type 2), which also ensures the resorption of 90–99% of the carnitine from the primary urine [78,79].

The effects of carnitine are being studied in obesity; athletes; pregnancy; ischemic heart disease; liver injury; and diseases of the nervous system, including depression, dementia, Alzheimer’s, and Parkinson’s diseases [80,81,82,83,84,85].

Disorders of carnitine synthesis due to TML-dioxygenase or butyrobetaine dioxygenase mutation are usually not associated with carnitine deficiency because most of it is obtained from the diet [86]. Carnitine transport defect due to the pathogenic variants of OCTN2 results in intracellular carnitine deficiency; impaired fatty acid oxidation; decreased glycemia due to preferred glucose oxidation; and fat accumulation, especially in the liver, muscles, and myocardium [86,87].

It has been shown that a moderate excess of dietary lysine lowers plasma and tissue carnitine concentrations in pigs [88]. Carnitine content increases in the liver more than twentyfold, and lysine concentration in plasma decreases in fenofibrate (a drug for hyperlipidemia therapy)-treated rats [89]. Increased levels of acylcarnitines are important biomarkers of fatty acid oxidation disorders [90].

### 5.5. Lysine Decarboxylation to Cadaverine

Cadaverine is a foul-smelling diamine formed by lysine decarboxylase expressed in bacteria but not in animal tissues (Figure 11). In humans, cadaverine is produced primarily by intestinal and oral microbiota, and together with some other polyamines, it is the cause of the smell of the stool and oral malodor [91]. Cadaverine activates, via trace amine-associated receptors (TAARs) in the olfactory epithelium, neural circuits that serve as signals evoking some types of stereotyped behavior [92]. In the oral cavity, increased levels of cadaverine arising from the putrefaction of debris demonstrated leukocyte migration disruption and a role in periodontitis development [93].

## 6. Lysine as a Dietary Supplement and Its Therapeutic Potential

There are several applications of lysine use as a dietary supplement; the potential application of the phenomenon of lysine–arginine antagonism (Section 3) is investigated in connection with the therapy of viral infections, disorders of immune and cardiovascular functions, and carcinogenesis. Lysine analogs have been found to have clinical use in preventing bleeding, and polylysine is promising in treating viral diseases (Figure 12).

### 6.1. Prophylaxis and Therapy of Lysine Deficiency

Lysine deficiency may occur in poor countries and groups of people with low socioeconomic status, where cereals are a staple food providing poor-quality protein, and in people who consume low-protein diets, such as vegetarians and patients with chronic renal failure. A marked lysine deficiency has been observed in children with kwashiorkor [94,95]. The loss of lysine during food processing can play an additional role in lysine deficiency [10,91]. The symptoms of lysine deficiency include decreased appetite, weight loss in adults, growth delay in children, anemia, fatigue, and mood changes, such as irritability and agitation.

Several articles have described the nutritional benefits of lysine oral supplements, grain legumes, and lysine-fortified diets for the therapy of kwashiorkor and proper development, primarily in children and adolescents, general well-being, and prevention of cardiometabolic disorders [1,2,8,94,95,96].

### 6.2. Muscle Performance

Lysine is a precursor of carnitine that is involved in fatty acid oxidation in mitochondria and, like arginine and some other amino acids, stimulates growth hormone release and its concentrations in plasma [97]. Moreover, lysine was shown to reduce skeletal muscle protein degradation [98]. Therefore, many individuals, primarily athletes, believe that lysine supplementation promotes muscle gain and strength [99,100].

Unfortunately, most published data do not support this assumption, and no properly conducted scientific studies have shown that oral lysine supplementation before strength training increases muscle performance to a greater extent than strength training alone [99]. Furthermore, lysine supplementation to a lysine-sufficient diet adversely affected the growth of rats [101]. In summary, the beneficial effects of lysine supplementation observed in people consuming a lysine-deficient diet do not appear in subjects who consume a diet with sufficient lysine content.

### 6.3. Herpes Simplex Infections Therapy

Because exogenous arginine is essential for forming capsid proteins of the herpes simplex virus and its multiplication, the displacement of arginine by L-lysine is used to treat herpetic infections (cold sores). The therapeutic forms include an ointment and L-lysine hydrochloride orally in 1–3 g/day dosages. Unfortunately, a recent literature review uncovered no convincing evidence that lysine is effective in treating herpes simplex lesions [4].

### 6.4. Modulation of Immune and Cardiovascular Functions

Lysine levels are decreased in nascent metabolic syndrome and negatively correlate with cardio-metabolic features and inflammatory biomarkers [102]. It has been demonstrated that the interruption of arginine uptake by lysine administration may have therapeutic use in diseases characterized by NO overproduction, such as septic shock. For instance, preincubation of activated macrophages in 2 mM L-lysine reduced the intracellular L-arginine concentration from 2 mM to 160 µM and completely abolished NO synthase activity [25]. In another study, L-lysine blocked NO production and plasma nitrate and nitrite levels in rats with heart failure [103].

### 6.5. Cancer Therapy

Lysine is the most utilized essential amino acid in some types of cancer, whereas arginine has been proven to exhibit anti-cancer effects [3,104]. Therefore, the interruption of lysine intake by increased arginine levels can impact cell proliferation. The possibility is thoroughly investigated, but without clear conclusions [3,104,105]. Several articles have demonstrated that the dysregulation of lysine PTM, primarily acetylation, lactylation, sumoylation, neddylation, and crotonylation of histones, contributes to carcinogenesis [45,46,47,48,61,62,63,65]. Therefore, targeting specific writers and erasers of lysine PTM appears to be a promising strategy in the treatment of cancer.

### 6.6. Elimination of Harmful Substances

Lysine proved to inhibit the tubular reabsorption of positively charged proteins and peptides appearing in primary urine due to its binding to negatively charged proximal tubular cells. The effect of lysine was higher when compared with other positively charged amino acids, including ornithine, arginine, and ε-amino-caproic acid [106]. The phenomenon is used clinically to reduce renal uptake of harmful compounds and protect the kidneys from nephrotoxicity [107,108]. It has been shown that D-lysine may be preferred when compared to L-lysine due to its reduction in renal uptake of radioactivity during scintigraphy [109].

### 6.7. Osteoporosis Therapy

Animal and human studies demonstrated that lysine enhances intestinal and renal calcium absorption and the potential usefulness of L-lysine supplements for both preventive and therapeutic interventions in osteoporosis [110,111,112,113]. Studies performed under in vitro conditions have demonstrated that lysine increases osteoblast proliferation and differentiation and positively affects hydroxyapatite mineralization and bone mineral density [114,115]. Unfortunately, clinical studies confirming the link between lysine intake and osteoporosis protection do not exist.

### 6.8. Wound Healing

Lysine is essential for the synthesis of collagen and elastin; the formation of crosslinks with glutamine resistant to mechanical influences; and the stabilization of the extracellular matrix, hemostasis, and the activation of growth factors [50,56]. Hence, lysine has been suggested in the form of creams, gels, and sprays to support wound healing. Beneficial effects of lysine and lysine hyaluronate have been reported in the management of diabetic foot ulcers, hospitalized patients with decubitus ulcers, and chemo/radiotherapy-induced oral mucositis [116,117,118].

### 6.9. Polylysine and Therapy of Viral Infections

Amino groups located both at the α-carbon and the ε-carbon of lysine can be used to form peptides called α-polylysine or ε-polylysine, which contain a positively charged hydrophilic amino group and can bind to negatively charged surfaces of cells or proteins. α-Polylysine is commonly used to coat culture materials as an adhesion factor; ε-polylysine exerts antibacterial activity and is used as a food preservative, primarily in Japan and Korea [119,120].

Because some viruses form noninfectious complexes with polylysine, nanomaterials derived from lysine have been investigated as an option to prevent the replication of viruses and their entry into the cells. Polylysine deposited on the surface of human T-lymphocytes proved to have an inhibitory influence on the replication of human immunodeficiency virus, respiratory syncytial virus, influenza A virus, herpes simplex virus type 1, human cytomegalovirus, and SARS [7,121].

### 6.10. Lysine Analogs and Prevention of Bleeding

Tranexamic and ε-aminocaproic acid are lysine analogues with affinity to lysine binding sites of plasminogen to fibrin, resulting in inhibition of the conversion of plasminogen to plasmin that degrades the fibrin clots. These drugs are used as antifibrinolytic agents to reduce bleeding in trauma, surgery, cancer, and hematological disorders [5,6].

## 7. Adverse Effects of Increased Lysine Intake

The lysine dose associated with no observed adverse effect (NOAEL) in humans was determined at 6 g per day for long-term use [122]. Animal experiments have shown no signs of teratogenicity; a woman with hyperlysinemia bore a normal infant despite greatly increased lysine levels [122,123]. Most frequently mentioned adverse effects of increased lysine intake, primarily in the form of lysine hydrochloride, include symptoms of impaired gastrointestinal function, such as nausea, abdominal pain, and diarrhea. No reports suggested hyperchloremic acidosis.

However, several animal studies have demonstrated the detrimental effects of lysine on the kidneys. A high intravenous dose of lysine (600 mg/rat over 4 h) decreased glomerular filtration. It produced the morphological picture of acute tubular necrosis, whereas equivalent glycine, arginine, and glutamate doses produced no significant renal morphologic or functional changes [124]. Nephrotoxicity of lysine was also demonstrated in dogs infused with lysine hydrochloride [125]. The mechanism by which lysine may cause damage to renal tubules is not fully understood. Using a tubule epithelial cell line from a human kidney, it was demonstrated that lysine overloading induces apoptosis associated with activated NADPH oxidase signaling [126].

Several articles have demonstrated that excessive intake of a single amino acid can decrease the availability of other amino acids and significantly affect the whole body’s metabolism [127]. It has been shown that lysine supplementation dose-dependently reduced concentrations of all essential amino acids and adversely affected the growth of rats [101]. Some adverse effects of high intake of lysine can be due to the lysine–arginine antagonism phenomenon. Of clinical importance is the effect of an increased lysine-to-arginine ratio on NO production, vascular resistance, and tissue perfusion [25,103,128,129].

## 8. Summary and Conclusions

The results of studies referred to in this article demonstrate that lysine is an essential amino acid with a unique role in protein metabolism, nutrition, and epigenetic regulation. However, more work should be performed to obtain clear information on its physiological role and therapeutic potential.

First, it is necessary to reach conclusions about the importance of the pipecolate pathway of lysine catabolism in the brain and its role in neurometabolic disorders. Essential is a better understanding of the control of PTMs of lysine residues in proteins and their role in epigenetics. The challenge for biochemists and molecular biologists is to characterize the structure and function of writers, erasers, and readers of lysine PTMs. Further research is required to explore the role of lysine availability in homoarginine formation and how the phenomenon of lysine–arginine antagonism can be used to influence immune and cardiovascular systems and cancer. Of unique importance is the investigation of the therapeutic potential of lysine analogs and polylysine. Given the possible use of lysine to reduce the resorption of radionuclides and other harmful substances in the kidneys, its maximum dose without nephrotoxic effects must be determined.

In summary, there are still significant gaps in our knowledge of lysine metabolism and its role in various physiological and pathological conditions, and most of its potential therapeutic applications have not been unequivocally proven.

## Figures and Tables

**Figure 1 ijms-26-08791-f001:**
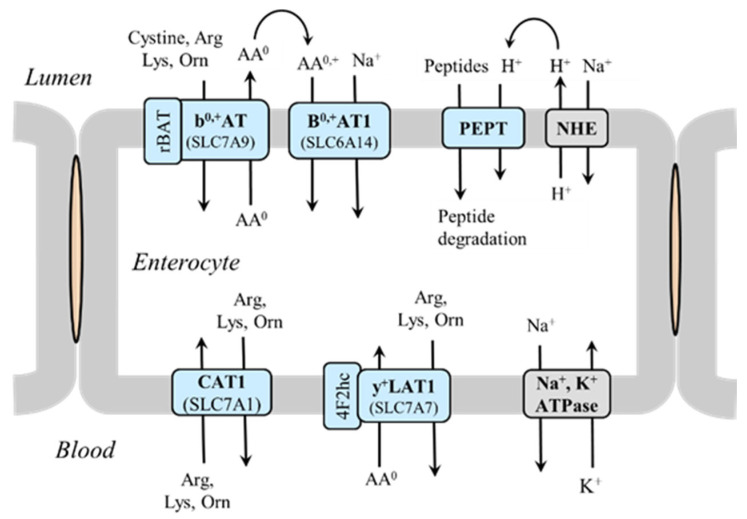
Absorption of cationic amino acids in the small intestine. In the apical side of enterocytes, transporter b^0,+^AT (SLC7A9) mediates the import of cationic amino acids and cystine in exchange for neutral amino acids (AA^0^), which are recaptured by Na^+^-dependent transporter B^0,+^AT1 (SLC6A14). CAT1 (SLC7A1) enables the diffusion of cationic amino acids down their electrogenic gradients, and y^+^LAT1 (SLC7A7) enables their efflux in exchange with neutral amino acids in the basolateral membrane. Abbreviations: AA^0^, neutral amino acids; AA^0,+^, neutral and basic amino acids; B^0,+^AT1, broad specificity amino acid transporter 1; NHE, Na^+^/H^+^ exchanger; PEPT, peptide transporter.

**Figure 2 ijms-26-08791-f002:**
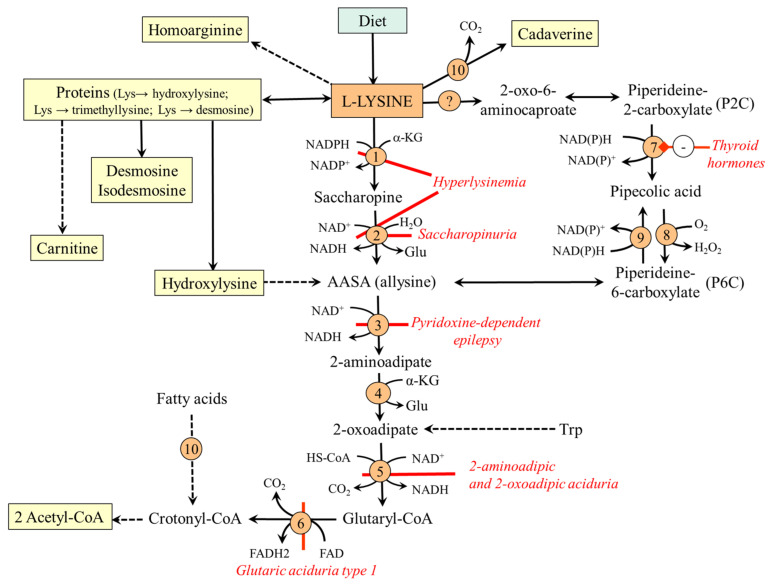
Main routes of lysine metabolism. 1, AASA synthase (lysine–ketoglutarate reductase domain); 2, AASA synthase (saccharopine dehydrogenase domain); 3, AASA dehydrogenase (antiquitin); 4, 2-aminoadipate transaminase; 5, 2-oxoadipate dehydrogenase; 6, glutaryl-CoA dehydrogenase; 7, ketimine reductase (µ-crystallin); 8, pipecolate oxidase; 9, pyrroline-5-carboxylate reductase; 10, β-oxidation. Abbreviations: AASA; 2-aminoadipate semialdehyde; P2C, piperideine-2-carboxylate; P6C, piperideine-6-carboxylate.

**Figure 3 ijms-26-08791-f003:**
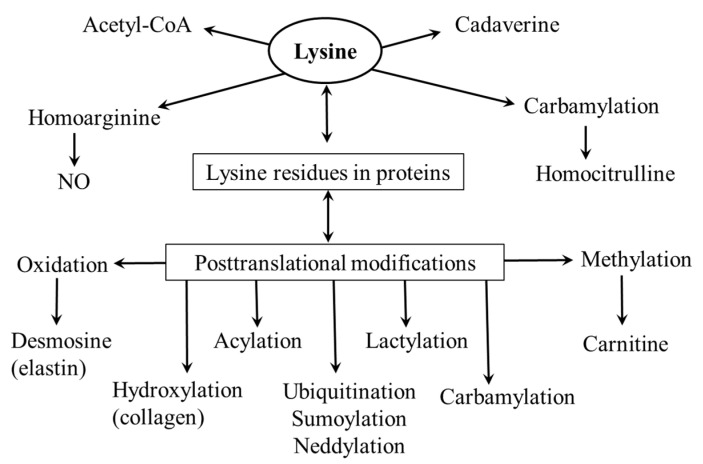
Physiological importance of lysine.

**Figure 4 ijms-26-08791-f004:**
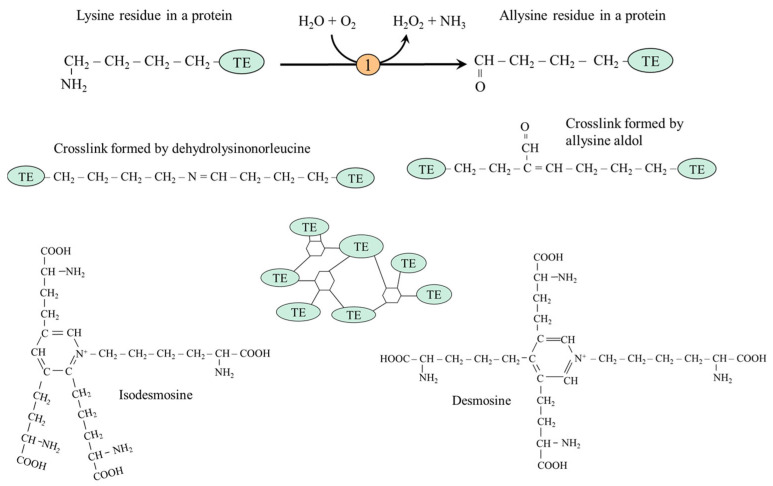
Formation of crosslinks among tropoelastins (TEs) in the elastin structure by dehydrolysinonorleucine, allysine aldol, isodesmosine, and desmosine. 1, lysyl oxidase.

**Figure 5 ijms-26-08791-f005:**
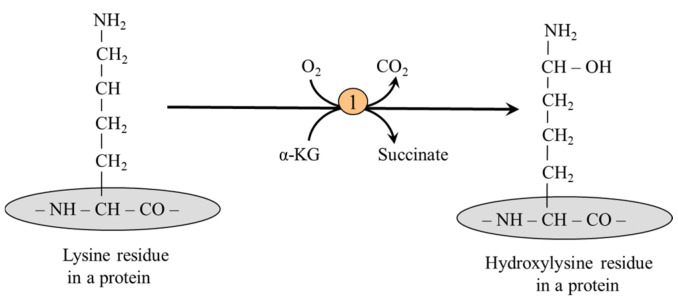
The formation of hydroxylysine in a collagen molecule. 1, lysyl hydroxylase.

**Figure 6 ijms-26-08791-f006:**
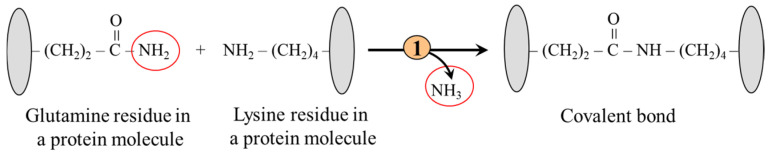
Covalent bond formation between glutamine and lysine residues. 1, transglutaminase.

**Figure 7 ijms-26-08791-f007:**
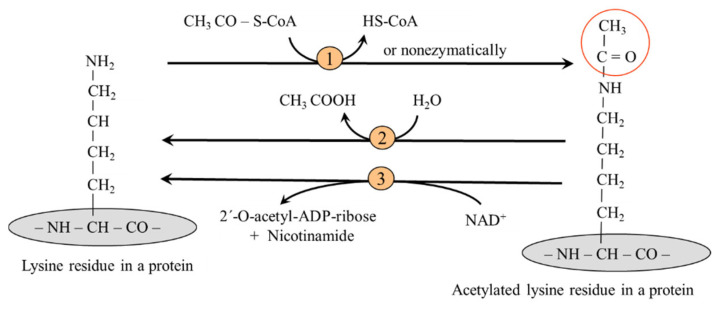
Acetylation and deacetylation of a lysine residue in a protein. 1, lysine acetyl transferase; 2, Zn^2+^-dependent lysine deacetylases; 3, NAD^+^-dependent lysine deacetylases (sirtuins).

**Figure 8 ijms-26-08791-f008:**
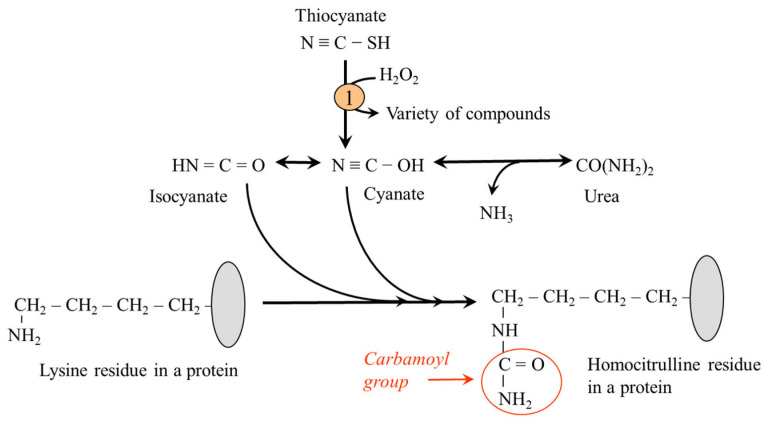
Lysine carbamylation. 1, myeloperoxidase.

**Figure 10 ijms-26-08791-f010:**
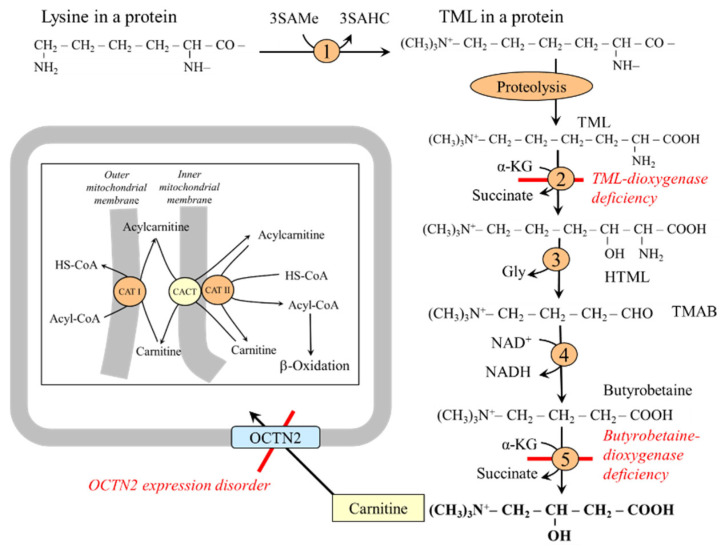
Role of lysine in carnitine synthesis. 1, protein lysine methyltransferase; 2, TML-dioxygenase; 3, HTML-aldolase; 4, TMAB-dehydrogenase; 5, butyrobetaine dioxygenase. Abbreviations: CACT, carnitine acylcarnitine translocase; CAT, carnitine acyltransferase; TML, N-trimethyllysine; HTML, 3-hydroxy-6-N-trimethyllysine; OCT, organic cationic transporter; TMABA, 4-N-trimethylaminobutyraldehyde; SAMe, S-adenosylmethionine; SAHC, S-adenosylhomocysteine.

**Figure 11 ijms-26-08791-f011:**
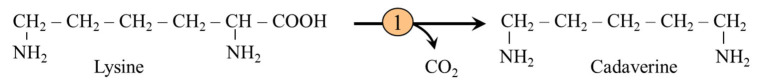
Cadaverine formation by lysine decarboxylation in bacteria. 1, lysine decarboxylase.

**Figure 12 ijms-26-08791-f012:**
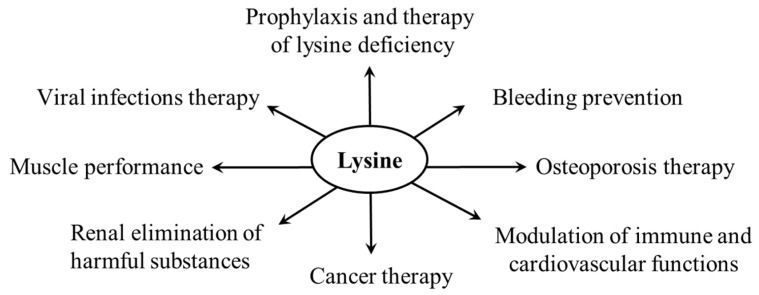
Supposed effects of lysine supplementation.

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
