# Peer review of "Lysine: Sources, Metabolism, Physiological Importance, and Use as a Supplement"

_ijms, 2025, doi:10.3390/ijms26188791_

Round 1

Reviewer 1 Report

Comments and Suggestions for Authors

In this review, the author provides a comprehensive summary of the physiological and pathological roles of lysine, both as a nutrient and as a component of proteins. The work will be valuable to other investigators as background information when developing their research interests. Nevertheless, I have a few concerns:

  • With regard to post-translational modifications, it would strengthen the manuscript to also discuss SUMOylation and neddylation.

  • Lysine has been reported to promote wound healing, for example in the treatment of diabetic foot ulcers, which could be included and discussed.

Comments on the Quality of English Language

While the overall quality of the language is acceptable, minor editing is recommended. For example, the phrasing on Page 2, lines 44–46, is somewhat unclear and would benefit from revision for clarity. 

Author Response

RESPONSE TO THE COMMENTS OF THE REVIEWER #1

Dear Reviewer,

thank you very much for your constructive remarks and positive evaluation of the article. Here are the responses to your comments:

Comment

With regard to post-translational modifications, it would strengthen the manuscript to also discuss SUMOylation and neddylation.

Response

Thank you very much for your suggestion. The sections devoted to lysine sumoylation and neddylation, including relevant references, have been added to the manuscript. Please see Page 9. In addition, we have added a brief section devoted to lysine lactylation (Page 10).

Comment

Lysine has been reported to promote wound healing, for example in the treatment of diabetic foot ulcers, which could be included and discussed.

Response

The following paragraph has been added to the manuscript:

6.8. Wound healing

Lysine is essential for the synthesis of collagen and elastin, the formation of cross-links with glutamine resistant to mechanical influences, and the stabilization of the extracellular matrix, hemostasis, and the activation of growth factors [50, 56]. Hence, lysine has been suggested in the form of creams, gels, and sprays to support wound healing. Beneficial effects of lysine and lysine hyaluronate have been reported in the management of diabetic foot ulcers, hospitalized patients with decubitus ulcers, and chemo/radiotherapy-induced oral mucositis [116-118].

Comments on the Quality of English Language

While the overall quality of the language is acceptable, minor editing is recommended. For example, the phrasing on Page 2, lines 44–46, is somewhat unclear and would benefit from revision for clarity.

Response

We have carefully checked the entire manuscript to improve its style and remove typos. A native speaker has checked the English. The written text's spelling, grammar, and tone were finally re-checked using the software tool Grammarly.

The inappropriate sentence "The unique chemical properties... of free lysine" on Page 2, lines 44-45, has been removed.

Milan Holeček

Reviewer 2 Report

Comments and Suggestions for Authors

Holecek's work is a review of the source and role of lysine in metabolism and the potential usefulness of its supplementation as a dietary supplement.

The work is well-written, clearly structured, and offers detailed and comprehensive biochemical analysis. However, in my opinion, the paragraphs in Section 6 require further exploration, particularly, but not limited to, Sections 6.2 and 6.5.

Furthermore, although Section 7 addresses the issue of the possible adverse effects of lysine supplementation, I encourage the author to consider and discuss the potential limitations of lysine supplementation alone (or any free-form AA) in altering the ratio of essential to nonessential amino acids. It is well known that alterations in this ratio, rather than individual AAs, often significantly affect overall cellular metabolism and tumor proliferation.

As has been done in Sections 3, 4, and 5, a figure of the possible "therapeutic" functions of lysine would be helpful.

Author Response

RESPONSE TO THE COMMENTS OF THE REVIEWER #2

Dear Reviewer,

thank you very much for your constructive remarks and positive evaluation of the article. Here are the responses to your comments:

Comment

However, in my opinion, the paragraphs in Section 6 require further exploration, particularly, but not limited to, Sections 6.2 and 6.5.

Response

Sections 6.2 and 6.5 have been rewritten as follows: 

6.2.        Muscle performance

Lysine is a precursor of carnitine that is involved in fatty acid oxidation in mitochondria and, like arginine and some other amino acids, stimulates growth hormone release and its concentrations in plasma [97]. Moreover, lysine was shown to reduce skeletal muscle protein degradation [98]. Therefore, many individuals, primarily athletes, believe that lysine supplementation promotes muscle gain and strength [99,100]. 

Unfortunately, most published data do not support this assumption, and no properly conducted scientific studies have shown that oral lysine supplementation before strength training increases muscle performance to a greater extent than strength training alone [99]. Furthermore, lysine supplementation to a lysine-sufficient diet adversely affected the growth of rats [101]. In summary, the beneficial effects of lysine supplementation observed in people consuming a lysine-deficient diet do not appear in subjects who consume a diet with sufficient lysine content. 

6.5.        Cancer therapy

Lysine is the most utilized essential amino acid in some types of cancer, whereas arginine has been proven to exhibit anti-cancer effects [3,104]. Therefore, the interruption of lysine intake by increased arginine levels can impact cell proliferation. The possibility is thoroughly investigated, but without clear conclusions [3,104,105]. Several articles have demonstrated that the dysregulation of lysine PTM, primarily acetylation, lactylation, sumoylation, neddylation, and crotonylation of histones, contributes to carcinogenesis [45-48, 61-63,65]. Therefore, targeting specific writers and erasers of lysine PTM appears to be a promising strategy in the treatment of cancer.

Comment

Furthermore, although Section 7 addresses the issue of the possible adverse effects of lysine supplementation, I encourage the author to consider and discuss the potential limitations of lysine supplementation alone (or any free-form AA) in altering the ratio of essential to nonessential amino acids. It is well known that alterations in this ratio, rather than individual AAs, often significantly affect overall cellular metabolism and tumor proliferation.

Response

The following sentences have been added to Section 7.

“Several articles have demonstrated that excessive intake of a single amino acid can decrease the availability of other amino acids and significantly affect the whole body's metabolism [127]. It has been shown that lysine supplementation dose-dependently reduced concentrations of all essential amino acids and adversely affected growth of rats [101].”

Comment

As has been done in Sections 3, 4, and 5, a figure of the possible "therapeutic" functions of lysine would be helpful.

Response

Done. Please, see Figure 12.

Milan Holeček
